# Comparing the effectiveness and safety of Abatacept and Tocilizumab in elderly patients with rheumatoid arthritis

**Jumpei Temmoku[1], Masayuki Miyata[2], Eiji Suzuki[3], Yuya Sumichika[1], Kenji Saito[1], Shuhei Yoshida[1], Haruki Matsumoto[1], Yuya Fujita[1], Naoki Matsuoka[1], Tomoyuki Asano[1], Shuzo Sato[1], Hiroshi Watanabe[1], Kiyoshi Migita[1]\***

**1** Department of Rheumatology, Fukushima Medical University School of Medicine, Fukushima, Fukushima, Japan, **2** Department of Rheumatology, Japanese Red Cross Fukushima Hospital, Fukushima, Japan, **3** Department of Rheumatology, Ohta Nishinouchi General Hospital Foundation, Koriyama, Fukushima, Japan

\* migita@fmu.ac.jp

## Abstract

### Background

The number of biological DMARDs (bDMARDs) used in elderly patients with rheumatoid arthritis (RA) has increased in recent years. We aimed to compare the drug retention rates and safety of abatacept (ABT) and tocilizumab (TCZ) in elderly patients with RA.

### Methods

A total 125 elderly patients with RA (>65 years) who began therapy with either ABT (n = 47) or TCZ (n = 78) between 2014 and 2021 at our institute were enrolled. We compared the drug retention rate and clinical response at 24 weeks between elderly patients with RA treated with ABT and those treated with TCZ. Adverse events (AEs) and the reasons for drug discontinuation were assessed.

### Results

There was no significant difference in demographic characteristics except for the use of glucocorticoid between the ABT and TCZ groups. There was no significant difference in the drug retention rate between the ABT and TCZ groups. Furthermore, there was no significant difference in the discontinuation rates due to the lack of effectiveness between these two groups. The proportions of the patients archiving low disease activity at 24 weeks did not differ significantly between the two groups. Whereas, the discontinuation rates due to AEs, including interstitial lung disease (ILD), seemed higher in the TCZ group than in the ABT group. In TCZ-treated group, the concomitant use of methotrexate (MTX) significantly increased the incidences of AEs leading to the discontinuation of TCZ. Whereas these was no significant impact of concomitant use of MTX on the incidences of AEs leading to discontinuation in ABT-treated group.

**Data Availability Statement:** All relevant data are within the manuscript and its Supporting Information files.

**Funding:** The author(s) received no specific funding for this work.

**Competing interests:** The authors have declared that no competing interests exist.

## Conclusions

In elderly patients with RA treated with ABT and TCZ, drug retention rates were equivalent between the two groups. There were some differences in safety profiles between ABT and TCZ, and the rates of discontinuation due to AEs, including ILD, seem to be lower with ABT than with TCZ in elderly patients with RA.

## Background

The prevalence of elderly patient with rheumatoid arthritis (RA) in elderly patients has been increasing, and the peak age at disease onset was found to be 60–69 years in JAPAN [1]. Elderly patients with RA seem to have characteristic clinical features and biological profiles that differ from those of younger patients with RA [2]. The management of elderly patients with RA seems challenging, since treatment approaches must be balanced against adverse events due to the many comorbidities associated with old age, including chronic lung disease and chronic kidney disease [3]. Elderly patients with RA who may have different RA pheno-types, it seems important to know which biological disease-modifying antirheumatic drugs (bDMARDs) can efficiently improve the RA disease activity in elderly patients [4].

There are several cohort studies comparing the effectiveness of bDMARDs among elderly patients with RA [5]. Abatacept (ABT) selectively inhibits T cell co-stimulatory signals, resulting in the suppression of inflammatory cytokines [6]. The incidence of serious infections seems lower among patients receiving abatacept than among those receiving other biological agents such as tumor necrosis factor inhibitors (TNFi) or interleukin-6 (IL-6) receptor antibodies [7]. Furthermore, real-world data have demonstrated that ABT treatment is associated with a lower risk of infections than treatment with other bDMARDs [8]. In contrast, significantly higher serum levels of IL-6 and lower serum levels of TNF-α were detected in elderly-onset rheumatic arthritis (EORA) than in young-onset RA, suggesting that IL-6 might be an important inflammatory mediator in elderly patients with RA [9].

Therefore, it is interesting to compare the effectiveness and safety of TCZ and ABT in elderly patients with RA who are intolerant to conventional synthetic disease-modifying anti-rheumatic drugs (csDMARDs). However, real-world data on the effectiveness and safety of TCZ and ABT in elderly patients with RA are limited. Given that drug retention rates can be influenced by both the safety and effectiveness of bDMARDs, we hypothesized that comparisons of these two bDMARDs are reasonable to estimate their usefulness in elderly patients with RA. From this viewpoint, this study aimed to investigate drug retention and reasons for discontinuation of ABT and TCZ among elderly patients with RA taking these two agents in an observational cohort study.

## Materials and methods

### Patients and study design

We conducted a retrospective cohort study at the Department of Rheumatology, Fukushima Medical University Hospital and Japanese Red Cross Fukushima hospital and Ohta Nishinou-chi Hospital. Among 1148 elderly patients (age ≥65 years; Female 796, Male 352) diagnosed with RA during the study period (between May, 2014 and July, 2021), 125 consecutive elderly (age ≥65 years) patients who were initiated with ABT or TCZ due to inadequate response to at least one or more csDMARDs were enrolled. All patients were diagnosed as having RA

according to the 2010 American College of Rheumatology (ACR) /European League Against Rheumatism (EULAR) classification criteria for RA [10] and be continuously followed up until 24 weeks after initiation of ABT or TCZ. The study was approved by the Institutional Review Board of Fukushima Medical University (No.29281), and Japanese Red Cross Fukushima Hospital and Ohta Nishinouchi Hospital.

## Clinical evaluations

At the start of treatment, baseline data were collected from medical records, including demographics, clinical data (disease duration, presence of the anti-citrullinated protein/peptide antibody [ACPA] antibody and rheumatoid factor [RF]), evaluations of disease activity (swollen joint count [SJC], tender joint count [TJC], patient global assessment [PtGA], physician global assessment [PGA]), and C-reactive protein [CRP], and information of treatments (current glucocorticoid and MTX doses, previous use of bDMARDs). Treatment was at the discretion of the attending physician, based on the clinical conditions and patient's intentions. Disease activity were evaluated based on disease Activity Score-28 (DAS28)-CRP and clinical disease activity index (CDAI). We divided the patients into those at remission (DAS28-CRP < 2.6) and those in non-remission as follows, low disease activity (LDA): $2.6 \leq$ DAS28-CRP < 3.2, moderate disease activity (MDA): DAS28-CRP 3.2–5.1, high disease activity (HDA): DAS28-CRP > 5.1 [11]. With respect to the CDAI disease activity categories, HDA is defined as a CDAI >22, MDA as a CDAI >10 and $\leq$22, LDA as a CDAI $\leq$10 and >2.8, and remission as a CDAI $\leq$2.8 [12].

## Follow-up

Serial assessments of disease activity including laboratory and treatment-related information were collected at every 4 weeks after the initiation of ABT and TCZ. If treatment was discontinued, the reasons for discontinuations were recorded. Clinical response was evaluated based on disease activity after 24 weeks from the start of ABT and TCZ. We compared the proportions of archiving LDA or remission between elderly patients initiating with ABT or TCZ [13]. To evaluate the appropriate therapeutic effect of ABT and TCZ, patients who had discontinued due to adverse events until 24 weeks were excluded for the evaluation of clinical response, and those who discontinued due to inadequate response were classified as HDA cases. AEs that had caused ABT or TCZ discontinuation were record in detail. Decisions to discontinue of ABT and TCZ due to AEs were determined carefully by 2 rheumatologists based on the evaluation of clinical findings, laboratory data and radiological examinations. If there were obvious causal relationships between AEs and the use of these bDMARDs, the discontinuation was decided.

## Statistical analysis

Continuous variables were showed as mean ± standard deviation or median (interquartile range) and categorical variables were sowed as frequency (percentage). The chi-squared test was used to compare categorical variables, and the Mann-Whitney U test was used to compare continuous variables. Drug retention was analyzed using Kaplan–Meier plots and assessed using the log-rank test. Cumulative incidences of discontinuation due to lack of effectiveness or adverse effects were compared using the Log-rank test for the Kaplan-Meier model. Statistical analyses were performed using the software of SPSS Statistics (version 25.0 for Windows, Chicago, IL, USA). Two–tailed p values <0.05 were considered indicative of statistical significance.

## Results

### Baseline characteristics of elderly RA patients

Among 1148 elderly (>65 years) RA patients registered in our cohort study, 125 (10.9%) who were on either ABT (n = 47) or TCZ (n = 78) during the observational period were enrolled (Fig 1). Relatively few patients were treated with ABT. The characteristics of the patients in each group are shown in Table 1. The proportion of patients treated with glucocorticoids (GCs) was significantly lower in the TCZ group than in the ABT group. Otherwise, there was no significant between-group difference between the two groups with respect to the other variables.

### Drug effectiveness at 24 weeks of follow-up

Proportions of patients who archived LDA or remission based on a DAS28-CRP or CDAI after 24 weeks were compared between elderly RA patterns initiated with ABT and TCZ. The proportion of disease activity categorized as remission, LDA, MDA and HDA at 24 weeks after initiation ABT or TCZ were shown in Fig 2. On week 24, there was no significant difference in the proportions of patients who achieved LDA or remission between the two groups (DAS28-CRP; ABT; 73.9% vs. TCZ; 80.5%, $p$ = 0.396, CDAI; ABT; 69.6% vs. TCZ; 66.7%, $p$ = 0.742). In the AMPLE study, the efficacy of ABT was shown to be higher in patients with RA with high titers of the ACPA [14]. Therefore, we compared the proportions of patients

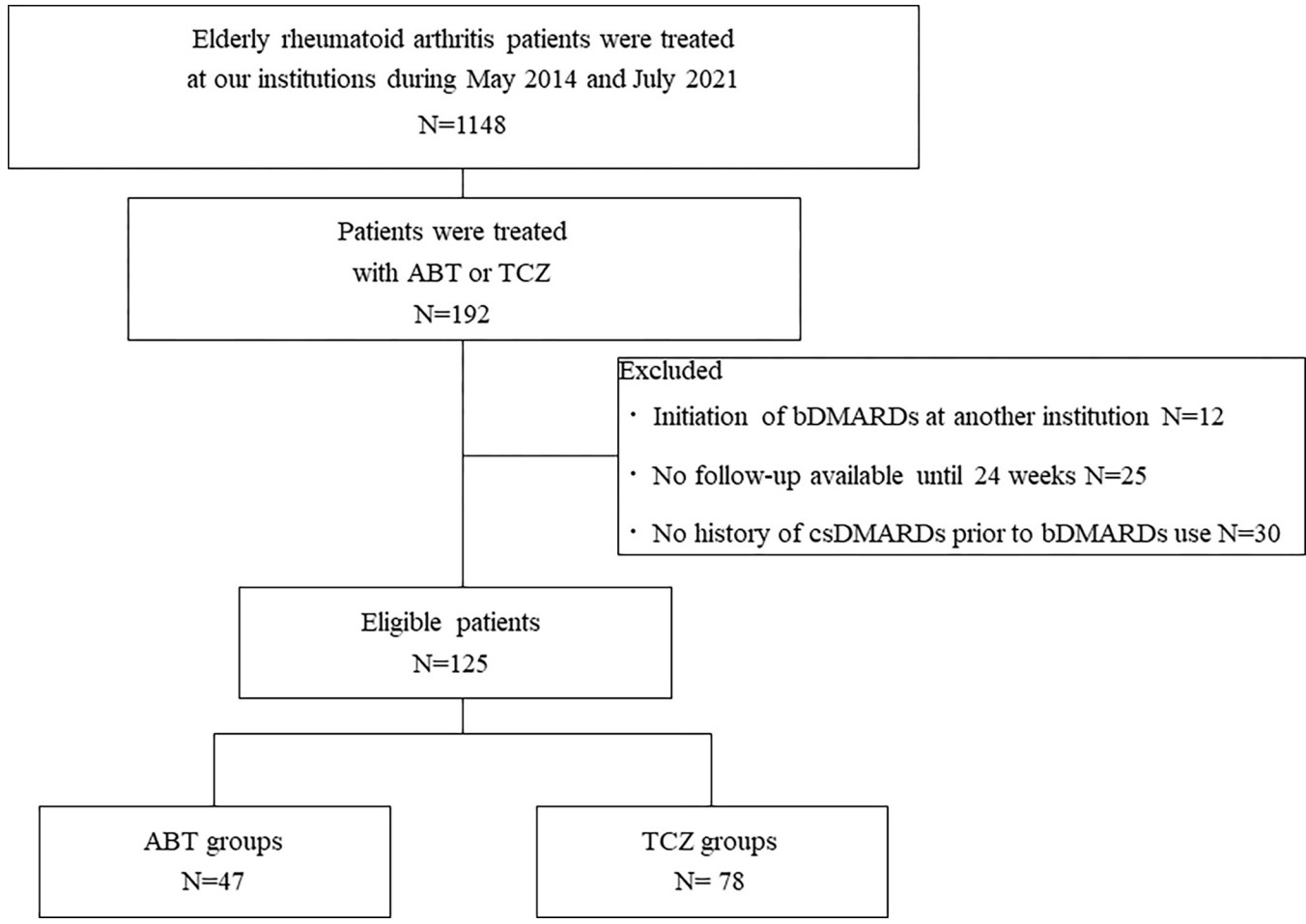

**Fig 1. Flowchart of patient recruitment in our cohort.**

**Table 1. Baseline characteristics between ABT group and TCZ group in elderly RA patients.**

|  | ABT (n = 47) | TCZ (n = 78) | p-Value |
|---|---|---|---|
| Age (years), median (IQR) | 74 (68–78) | 75 (70–81) | 0.220 |
| Female, n (%) | 37 (78.7) | 54 (69.2) | 0.172 |
| Intravenous/subcutaneous injection, n (%) | 36 (76.6)/11 (23.4) | 34 (43.6)/44 (56.4) | <0.001 |
| Disease duration (years), median (IQR) | 7 (0.9–15.0) | 7 (1.0–14.0) | 0.919 |
| RF-positive, n (%) | 39 (83.0) | 55 (70.5) | 0.092 |
| ACPA-positive, n (%) | 35 (74.5) | 60 (76.9) | 0.861 |
| CRP (mg/dL), median (IQR) | 1.93 (0.78–4.30) | 1.17 (0.29–3.88) | 0.289 |
| SJC, median (IQR) | 3 (1.5–5.5) | 2 (0.0–6.0) | 0.407 |
| TJC, median (IQR) | 2 (0.5–4.5) | 3 (1.0–8.0) | 0.204 |
| PtGA (mm), median (IQR) | 56.0 (30.5–80.0) | 49.0 (31.8–66.3) | 0.478 |
| PGA (mm), median (IQR) | 60.0 (33.5–72.5) | 48.0 (37.0–58.0) | 0.202 |
| DAS28-CRP, median (IQR) | 3.79 (2.80–4.29) | 3.52 (2.45–4.76) | 0.828 |
| CDAI, median (IQR) | 16.7 (8.6–31.0) | 16.7 (7.6–26.0) | 0.802 |
| eGFR (mL/min), median (IQR) | 71.0 (57.7–82.5) | 75.5 (60.9–84.1) | 0.522 |
| CKD (eGFR<60), n (%) | 13 (27.7) | 19 (24.4) | 0.732 |
| Interstitial lung disease, n (%) | 14 (29.8) | 17 (21.8) | 0.480 |
| MTX use, n (%) | 22(46.8) | 31 (39.7) | 0.439 |
| MTX dose (mg/week) | 6.0 (6.0–8.0) | 7.0 (4.0–8.0) | 0.941 |
| Other csDEMARDs, n (%) | 21 (44.7) | 22 (28.2) | 0.060 |
| GC use, n (%) | 29 (61.7) | 32 (41.0) | 0.025* |
| GC dose (mg/day) | 5.0 (2.5–7.5) | 5.0 (4.0–7.9) | 0.371 |
| Prior bDMARDs use, n (%) | 12 (25.5) | 25 (32.0) | 0.439 |
| Prior TNFi use, n (%) | 11 (23.4) | 25 (32.0) | 0.301 |
| Follow up periods (month), median (IQR) | 28 (10–42) | 24 (12–45) | 0.661 |

ABT: abatacept; TCZ: tocilizumab; RA: rheumatoid arthritis; IQR: interquartile range; RF: rheumatoid factor; ACPA: anti-citrullinated peptide antibody; CRP: c-reactive protein; SJC: swollen joint count; TJC: tender joint count; PtGA: patient global assessment; PGA: physican global assessment; DAS28: disease activity score 28; CDAI: clinical disease activity index; eGFR: estimated glomerular filtration rate; CKD: chronic kidney disease; csDMARDs: conventional synthetic disease modifying anti-rheumatic drugs; MTX: methotrexate; GC: glucocorticoid; bDMARDs: biological disease modifying anti-rheumatic drugs; TNFi: tumor necrosis factor inhibitors

* $p<0.05$.

who achieved LDA or remission between elderly patients with RA with ACPA (>4.5 U/ml) or without ACPA in ABT-treated group. There was no significant difference in the proportions of patients who achieved LDA or remission between elderly patients with and without the positivity of ACPA in ABT-treated group (Fig 3A). Similarly, there was no significant difference in the proportions of patients who achieved LDA or remission between elderly patients with RA with or without the positivity of ACPA in TCZ-treated group (Fig 3A). In addition, we compared the proportions of patients achieving LDA or remission in the ABT and TCZ-treated group for elderly RA patients according to the use of MTX. There was no significant difference in the proportions of patients who achieved LDA or remission between elderly patients with and without MTX in ABT and TCZ-treated group (Fig 3B).

## Drug overall retention rates

The overall drug retention rates of ABT and TCZ are shown in Fig 4. In ABT group, the rates of reasons for drug discontinuation were lack of efficacy 12 (25.5%) and AEs 6 (12.8%). In

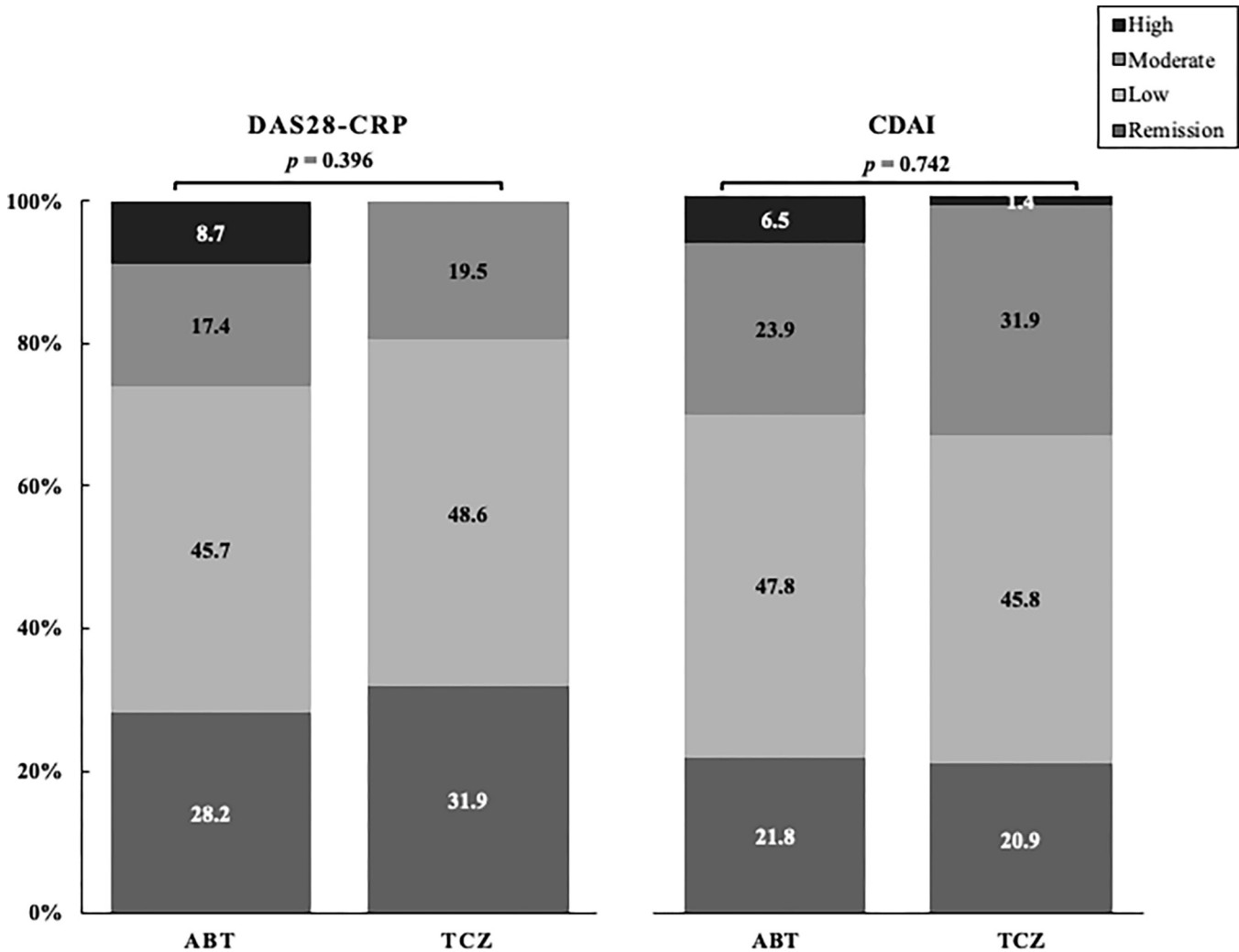

**Fig 2. Proportions of patients who archived LDA or remission based on a DAS28-CRP or CDAI between elderly patients initiating with ABT and TCZ at 24 weeks.** Disease activity was classified into four categories as HDA, MDA, LDA, remission based on DAS28-CRP and CDAI. There was no significant difference in the proportions of archiving LDA or remission between elderly patients initiating with ABT or TCZ.

TCZ group, the rates of reasons for drug discontinuation were lack of efficacy 14 (17.9%) and AEs 17 (21.8%). There was no significant difference in the drug retention rates between the two groups. The cumulative incidence of drug discontinuation due to lack of effectiveness was also compared between the two groups, and no significant difference in this parameter was found between the two groups (Fig 5).

## Rates of discontinuation due to AEs

Among the 47 patients who initiated with ABT, 6 (12.8%) discontinued the drug due to AEs. Among the 78 patients initiated with TCZ, 17 (21.8%) discontinued the drug due to AEs. The patients experienced no severe or life-threatening AEs in both groups. The total rates of discontinuation due to AEs were higher in the TCZ group than in the ABT group in elderly patients with RA; however, there was no significant difference between these two groups (Table 2). These AEs leading to the discontinuation of bDMARDs included intestinal lung disease (ILD, n = 5), infections (n = 5) hematological disorders (n = 3), malignancy (n = 4),

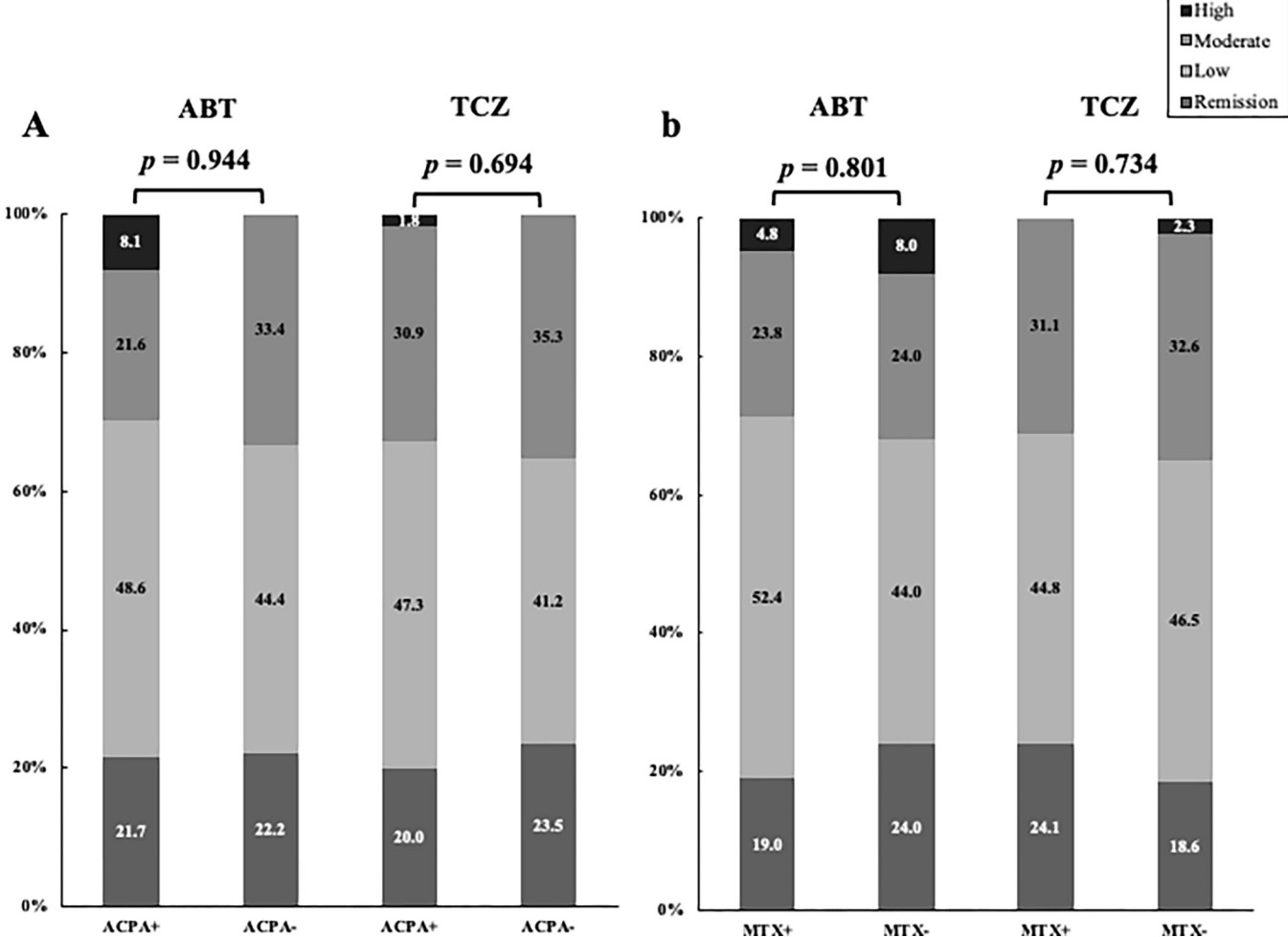

**Fig 3. Proportions of patients who archived LDA or remission based on CDAI between elderly patients initiating with ABT or TCZ at 24 weeks according to the presence of ACPA or the concomitant use of MTX.** Disease activity was classified into four categories as HDA, MDA, LDA, remission based on CDAI. (A) There was no significant difference in the portions of archiving LDA or remission between elderly patients initiating with ABT or TCZ according to the presence of ACPA. (B) There was no significant difference in the portions of archiving LDA or remission between elderly patients initiating with ABT or TCZ according to the use of MTX.

allergic reactions (n = 2), renal impairment (n = 1), liver dysfunction (n = 1), and cardiovascular event (n = 1). The most frequent AE leading to the discontinuation of TCZ in elderly RA patients was ILD. Among 5 patents complicated with ILD in TCZ group, 2 patients presented with the exacerbation of pre-existing ILD and 4 patients with the concomitant use of MTX. In contrast, ILD was not observed as an AE leading to the discontinuation of ABT in elderly RA patients. We compared the incidence of discontinuation due to AEs between the ABT and TCZ groups using the Kaplan-Meier curve. The incidence of discontinuation due to AEs was higher in the TCZ group than in the ABT group; however, there was no statistically significant difference in this parameter between the two groups (Fig 6).

## Rates of discontinuation due to AEs according to the use of MTX

We compared the demographic data between elderly RA patients who did and did not discontinue bDMARDs due to AEs (Table 3). We found that the use of MTX seems to be higher in

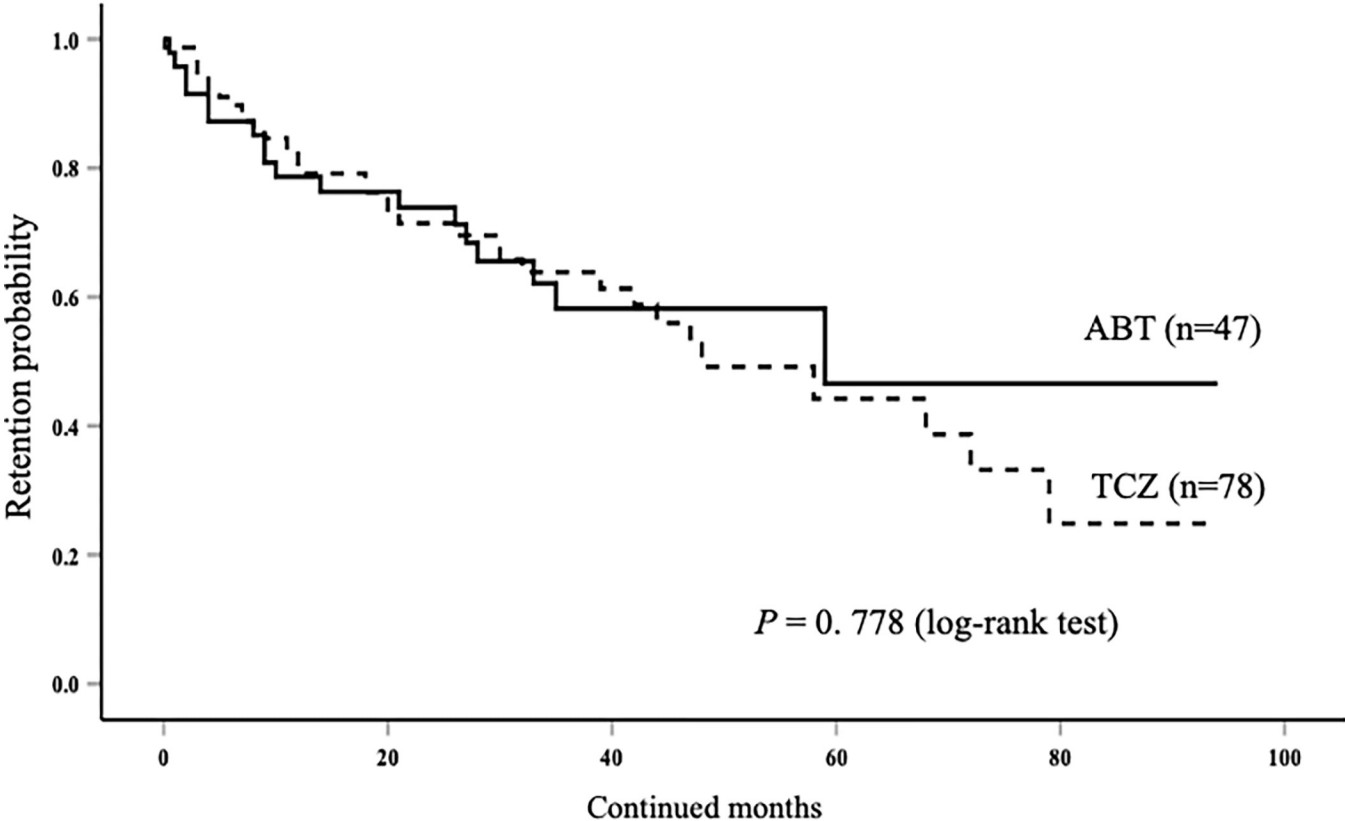

**Fig 4. Kaplan–Meier curve related to the overall cumulative drug retention rate of ABT and TCZ in elderly patients initiating these bDMARDs.** There was no significant between-group difference with respect to the drug retention rates.

patients discontinued bDMARDs due to AEs, however, there was no significant difference. The rates of combination use of MTX plus TCZ were significantly higher in patients who discontinued bDMARDs due to AE. We compared the incidence of discontinuation due to AEs according to the use of MTX using the Kaplan-Meier curve. As show in Fig 7A, the incidences of discontinuation of TCZ due to AEs were significantly higher in elderly RA patients with the use of MTX compared to those without use of MTX. Whereas there was no significant difference in the incidences of discontinuation of ABT between elderly RA patients with or without use of MTX (Fig 7B). We also compared the rates of each adverse event leading to the discontinuation of bDMARDs according to the use of MTX. In ABT group, there was no significant difference in the rates of each adverse events between elderly RA patients with and without use of MTX. In TCZ group, the rates of ILD were higher in elderly RA patients with the use of MTX compared to those without use of MTX, however, there was no statistical difference.

Finally, we performed the same analysis subjected the selected EORA (more than 60 years) RA patients (n = 85). Similarly, there was no significant difference in the overall cumulative drug retention rates of ABT and TCZ in these EORA patients initialing these biologics (Fig 8). The rates of discontinuation of these biologics due to AEs were significantly higher in TCZ group compared to those in ABT group (Fig 9). There was no difference in the rates of ABT discontinuation due to AEs between these EORA patients with or without use of MTX. Whereas, the rates of TCZ discontinuation due to AEs were significantly higher in these EORA patients with the use of MTX compared to those without the use of MTX (Fig 10).

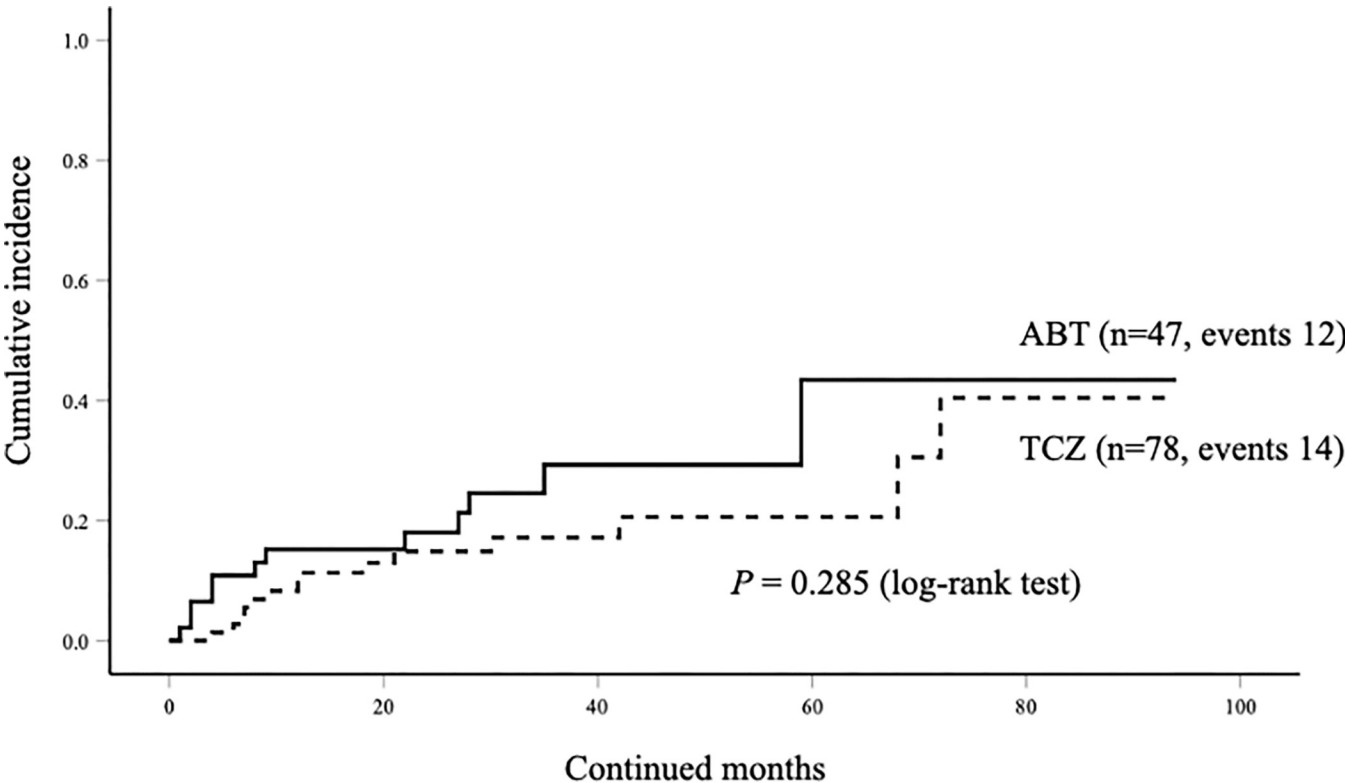

**Fig 5. The cumulative incidences of discontinuation of ABT or TCZ due to lack of effectiveness.** There were no significant differences in the incidences of drug discontinuation due to lack of effectiveness between ABT and TCZ groups.

## Discussion

The proportion of elderly patients with RA has increased, and these patients often have multiple comorbidities, which makes it difficult to treat them using bDMARDs [15]. In clinical trials, ABT has shown similar efficacy to other bDMARDs [16]. Furthermore, ABT treatment has been shown to be is efficacious in the treatment of elder patients with RA [17]. In contrast,

**Table 2. All adverse events and adverse events lead to drug discontinuation in ABT group and TCZ group.**

|  | AEs with drug discontinuation (Whole AEs) | | |
|---|---|---|---|
|  | ABT (n = 47) | TCZ (n = 78) | *p*-Value |
| Intestinal lung disease | 0 (0) | 5 (5) | 0.090 |
| Infection | 2 (9) | 3 (8) | 0.910 |
| Malignancy | 1 (1) | 3 (3) | 0.516 |
| Renal impairment | 1 (1) | 0 (0) | 0.376 |
| Liver dysfunction | 0 (0) | 1 (2) | 0.624 |
| Hematological disorder | 1 (3) | 2 (3) | 0.684 |
| Allergic reaction | 0 (0) | 2 (2) | 0.387 |
| Cardiovascular event | 0 (0) | 1 (2) | 0.624 |
| Hypothyroidism | 1 (1) | 0 (0) | 0.376 |
| Total | 6 (15) | 17 (25) | 0.207 |

AEs: adverse events; ABT: abatacept; TCZ: tocilizumab

* $p < 0.05$.

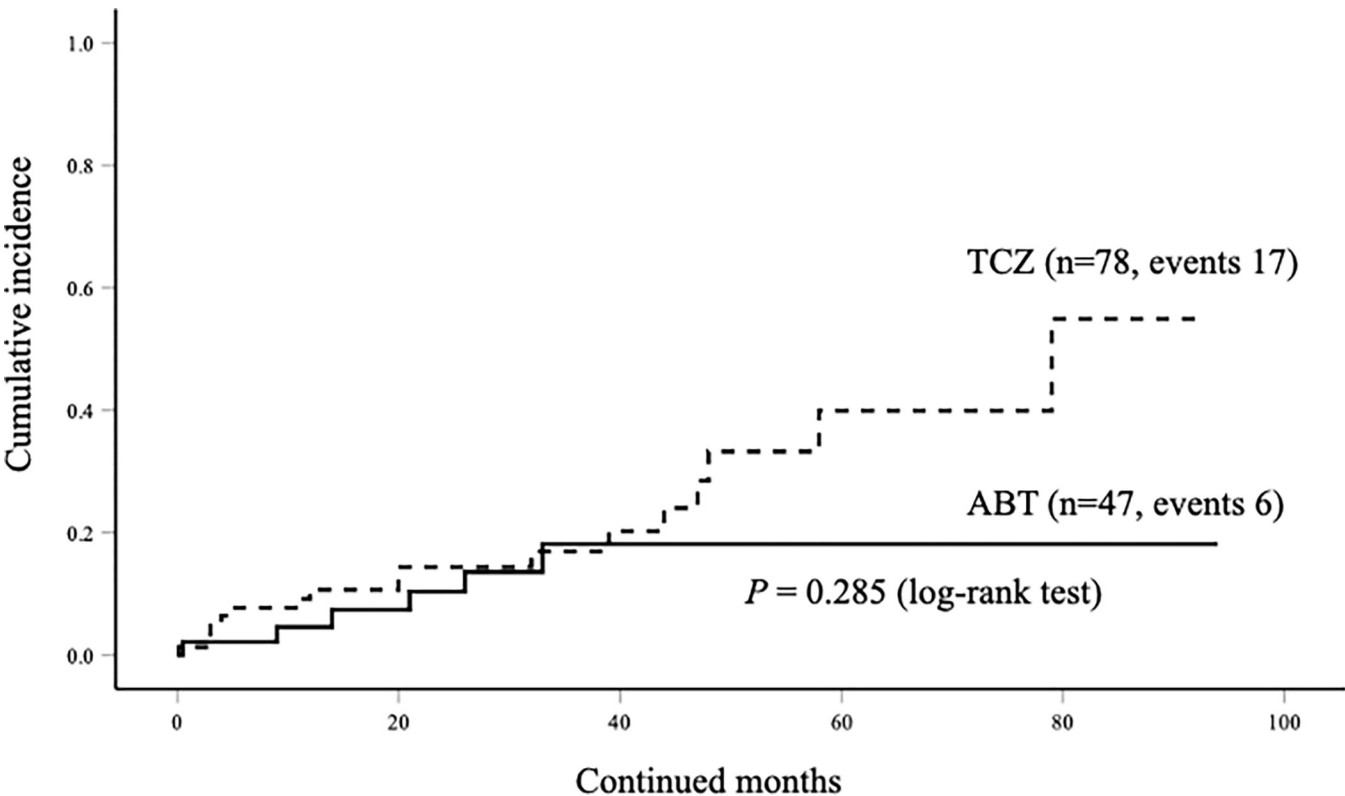

**Fig 6. The cumulative incidences of discontinuation of ABT or TCZ due to adverse.** There were no significant differences in the incidences of drug discontinuation due to AEs between ABT and TCZ groups.

**Table 3. Baseline characteristics of elderly RA patients with the discontinuation of bDMARDs due to AEs.**

| | AEs (n = 23) | Non-AEs (n = 102) | *p*-Value |
|---|---|---|---|
| Age (years), median (IQR) | 70 (66–77) | 73 (68–78) | 0.531 |
| Female, n (%) | 15 (65.2) | 76 (74.5) | 0.366 |
| Disease duration (years), median (IQR) | 7 (3–12) | 7 (1–16) | 0.958 |
| RF-positive, n (%) | 19 (82.6) | 77 (75.5) | 0.521 |
| ACPA-positive, n (%) | 14 (60.9) | 80 (78.4) | 0.145 |
| CRP (mg/dL), median (IQR) | 2.11 (0.76–3.85) | 1.43 (0.49–3.95) | 0.461 |
| DAS28-CRP, median (IQR) | 3.12 (2.67–3.72) | 3.89 (2.58–4.80) | 0.370 |
| CDAI, median (IQR) | 16.7 (9.4–26.0) | 16.8 (12.0–42.4) | 0.787 |
| eGFR (mL/min), median (IQR) | 60.0 (46.0–83.0) | 72.4 (61.5–82.6) | 0.157 |
| Interstitial lung disease, n (%) | 7 (30.4) | 24 (23.5) | 0.505 |
| bDMARDs use | | | |
| ABT, n (%) | 6 (26.1) | 41 (40.2) | 0.207 |
| TCZ, n (%) | 17 (73.9) | 61 (59.8) | 0.207 |
| ABT/MTX, n (%) | 3 (13.0) | 19 (18.6) | 0.525 |
| TCZ/MTX, n (%) | 11 (47.8) | 21 (20.6) | 0.007* |
| MTX use, n (%) | 14 (60.8) | 40 (39.2) | 0.058 |
| GC use, n (%) | 13 (56.5) | 48 (47.1) | 0.412 |

RA: rheumatoid arthritis; bDMARDs: biological disease modifying anti-rheumatic drugs; AEs: adverse events; IQR: interquartile range; RF: rheumatoid factor; ACPA: anti-citrullinated peptide antibody; CRP: c-reactive protein; DAS28: disease activity score 28; CDAI: clinical disease activity index; eGFR: estimated glomerular filtration rate; ABT: abatacept; TCZ: tocilizumab; MTX: methotrexate; GC: glucocorticoid

* $p < 0.05$.

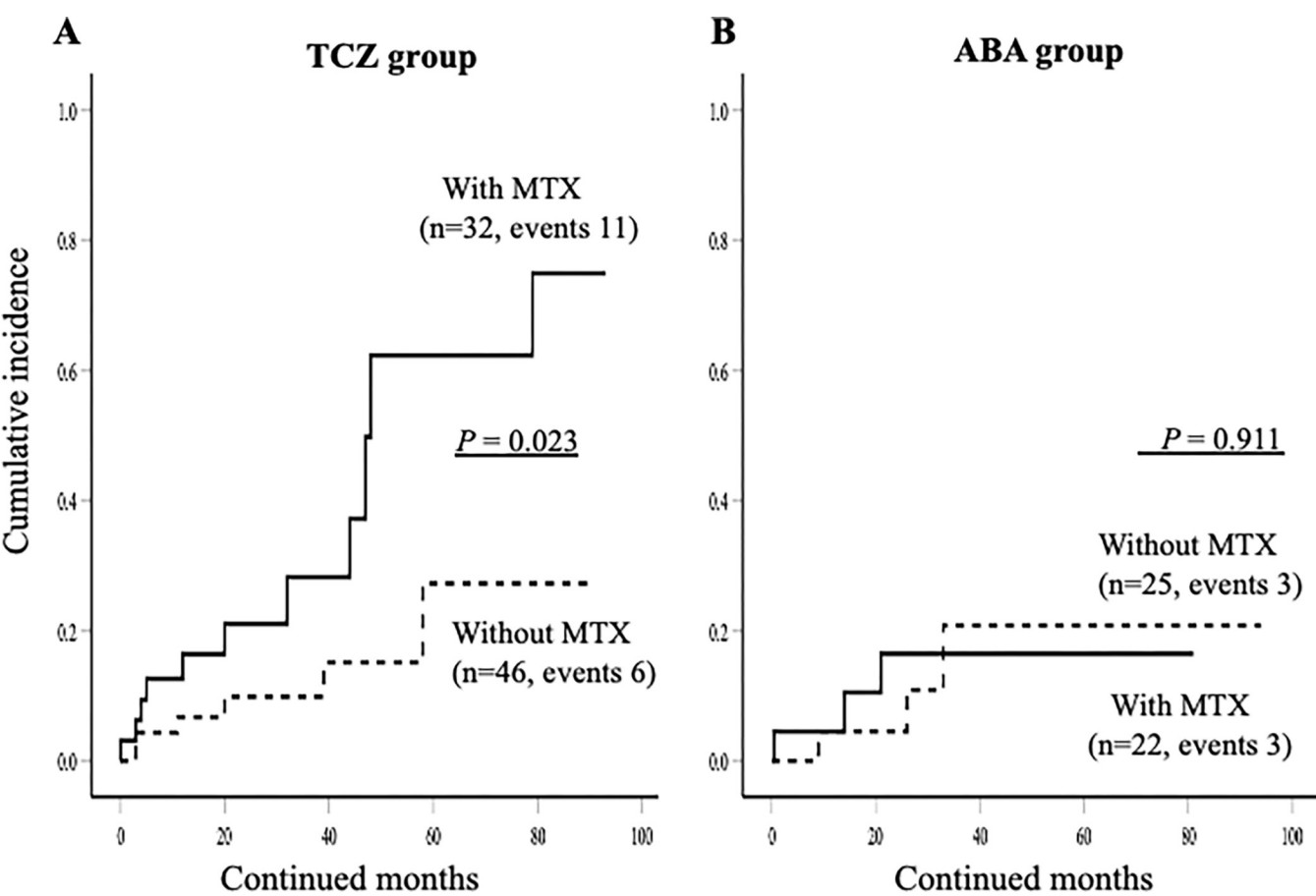

**Fig 7. The cumulative incidence of the discontinuation of ABT or TCZ due to AEs were compared in elderly RA patients group according to the concurrent MTX use.** (A) The incidences of the discontinuation of TCZ due to AEs were significantly higher in elderly RA patients with the use of MTX compared to those without the use of MTX. (B) There were no significant differences in the incidences of the discontinuation of ABT due to AEs between elderly RA patients group according to the concurrent MTX use.

previous reports have demonstrated that elderly patients with RA have higher serum levels of interleukin (IL-6), suggesting that IL-6-targeting therapy could be one of the viable therapeutic options in elderly patients with RA [18]. However, elderly patients may not be likely to participate in clinical trials of bDMARDs, which explains the scarcity of evidence of the effectiveness of TCZ or ABT in the elderly patients with RA. In this observational study involving the elderly (aged >65 years) patients with RA, we compared the effectiveness and safety of ABT and TCZ in the elderly patients with RA. Our study demonstrated the equivalent drug relation rates of ABT and TCZ in elderly patients with RA on these bDMARDs. In terms of the disease activity, our findings demonstrated that the degree of clinical response was comparable between elderly patients with RA taking ABT and those taking TCZ. Biological factors, including seropositivity for autoantibodies, may affect the effectiveness of bDMARDs in patients with RA [19]. It has been demonstrated that the titers of the ACPA influence the effectiveness of ABT in clinical trials [14]. However, our data demonstrated that the effectiveness of ABT and TCZ groups were not significantly influenced by the ACPA positivity in elderly patients with RA.

Previous studies reported that the drug retention rates for TCZ and the rates of discontinuation due to AEs were shown to be similar between elderly (≧65 years) patients and non-elderly (<65 years) patients [20]. In our data, elderly patients with RA treated with TCZ

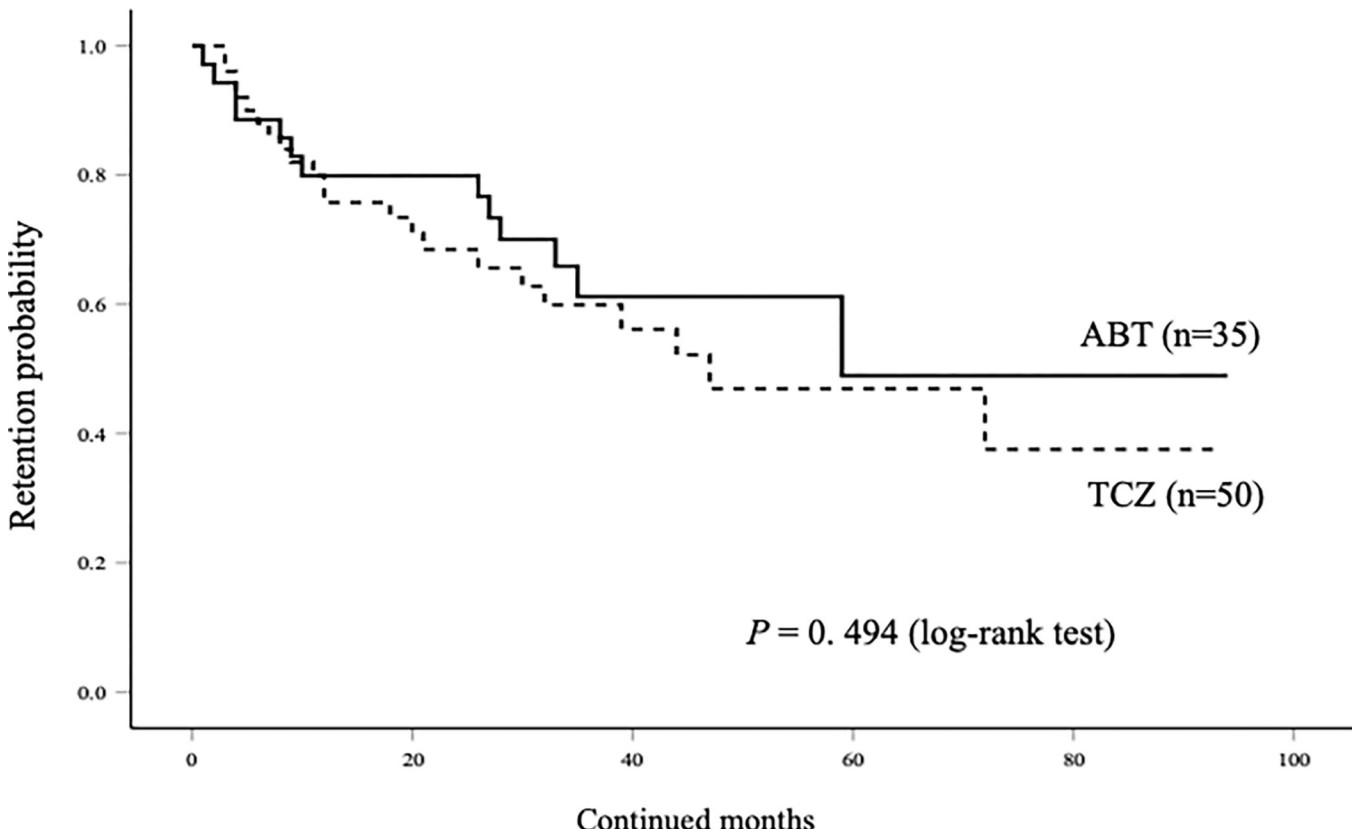

**Fig 8. Kaplan–Meier curve related to the overall cumulative drug retention rate of ABT and TCZ in EORA (more than 60 years) patients initiating these bDMARDs.** There was no significant between-group difference with respect to the drug retention rates.

experienced higher rates of discontinuation of TCZ due to AEs compared to those with ABT in our study. Our data do not allow us to give any causal explanation for the higher rates of discontinuation of TCZ due to AEs. In our data, elderly patients with RA receiving MTX tend to have higher rates of discontinuation of TCZ due to AEs. Among the AEs leading to the discontinuation of TCZ, ILD was the most frequent.

The risk of AEs may influence therapeutic decision-making, including the choice of bDMARDs [21]. Our data demonstrated the rates of discontinuation due to lack of effectiveness were comparable between elderly patients with RA initiated with ABT and TCZ. However, the rates of discontinuation due to AEs were relatively higher in the TCZ group than in the ABT group. Recent real-world data indicated that the rates for TCZ discontinuation due to the loss of efficacy and AEs were similar between TCZ monotherapy and combination therapy with MTX [22]. However, our data indicated that the rates of discontinuation due to AEs were significantly higher in elderly RA patients receiving TCZ plus MTX compared to those receiving TCZ monotherapy. In accord to our data, recent meta-analysis demonstrated that TCZ in combination with MTX is associated with a small but significantly increased risks with AEs [23]. It is possible that the comorbidities associated with elderly RA patients, may explain the high rates of discontinuation due to AEs in our patients receiving MTX and TCZ.

The bDMARDs have been proposed to be a potential triggering factor for the acceleration of pre-existing ILD [24]. Among bDMARDs, TNFi and TCZ have been more commonly implicated in the pathogenesis of RA-related ILD than ABT and RTX [25]. In line with these

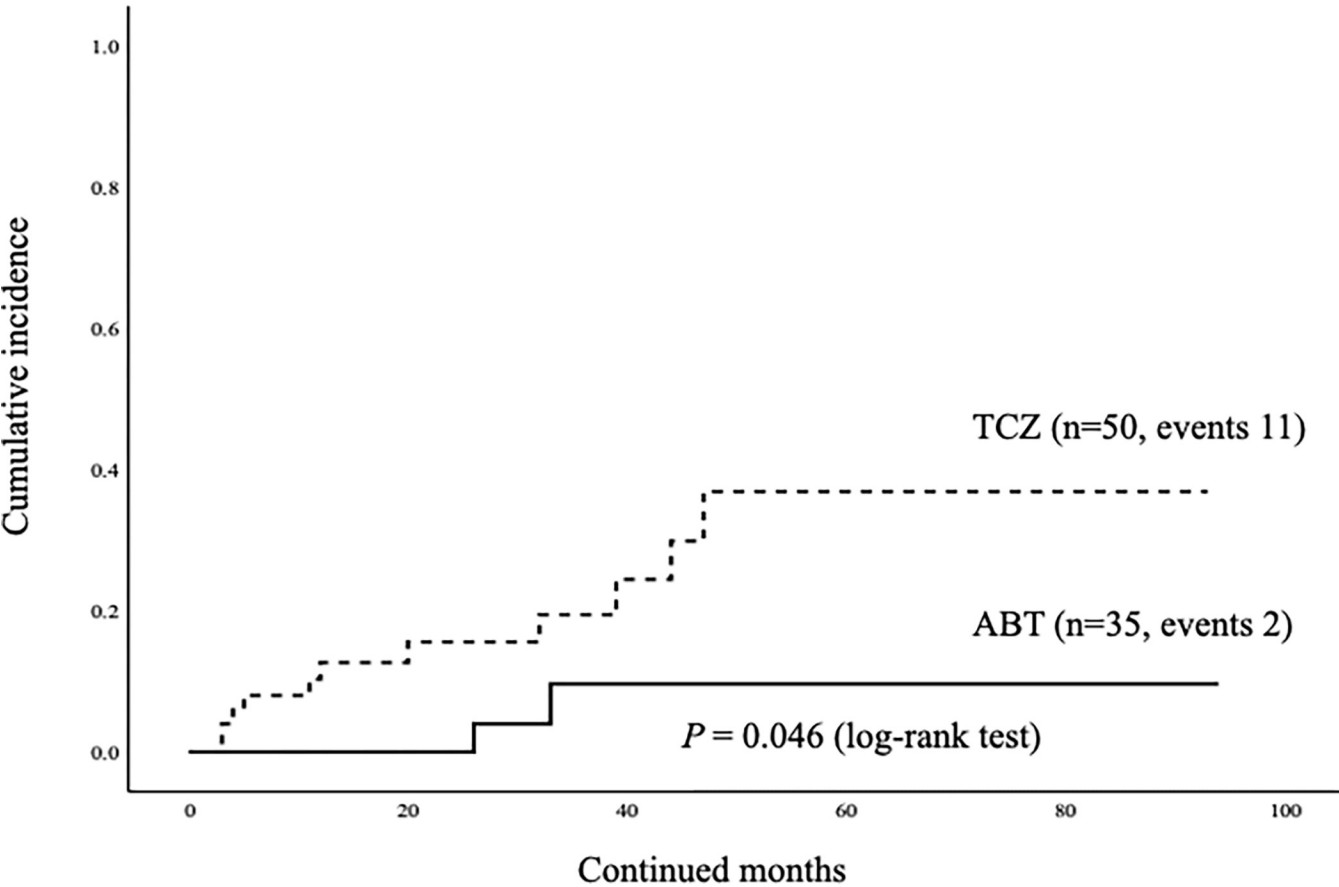

**Fig 9. The cumulative incidences of discontinuation of ABT or TCZ in EORA patients due to adverse.** The rates of discontinuation of these biologics due to AEs were significantly higher in TCZ group compared to those in ABT group.

reports, our data suggest that the use of ABT seems to be safe regarding the risk of worsening ILD in elderly RA patients. However, the precise mechanism by which ABT influences the ILD worsening in patients with RA has not yet been elucidated. Cytotoxic T lymphocyte-associated protein 4 (CTLA-4) blockers have been reported to induce immune-mediated pneumonitis [26]. The current evidence suggests the effectiveness and safety of ABT treatment in patients with RA with ILD [27]. A recent large-scale multicenter study on patients with RA with ILD demonstrated that ABT seems to be an effective and safe treatment option for these patients [28, 29]. Further studies are needed to determine whether ABT has a protective effect against RA-related ILD in elderly patients with RA.

Elderly patients with RA seem to have a characteristic pattern with a more acute onset, systemic involvement, and worse functional outcomes [30]. However, concerns about AEs may influence therapeutic decisions with clinicians often preferring a less aggressive approach in elderly patients [31]. Our data suggest that either ABT or TCZ could be an appropriate therapeutic choice for elderly patients with RA disease activity; however, risk-benefit profiles should be carefully assessed. Further large-scale clinical studies in elderly patients with RA are needed to elucidate the more detailed risk-benefit profiles of ABT and TCZ.

Nevertheless, our study has several limitations. First, this was a retrospective observational study, and the study design may have might have affected the evaluation of treatment effectiveness. Second, the choice of treatment and decision to discontinue were made at the discretion

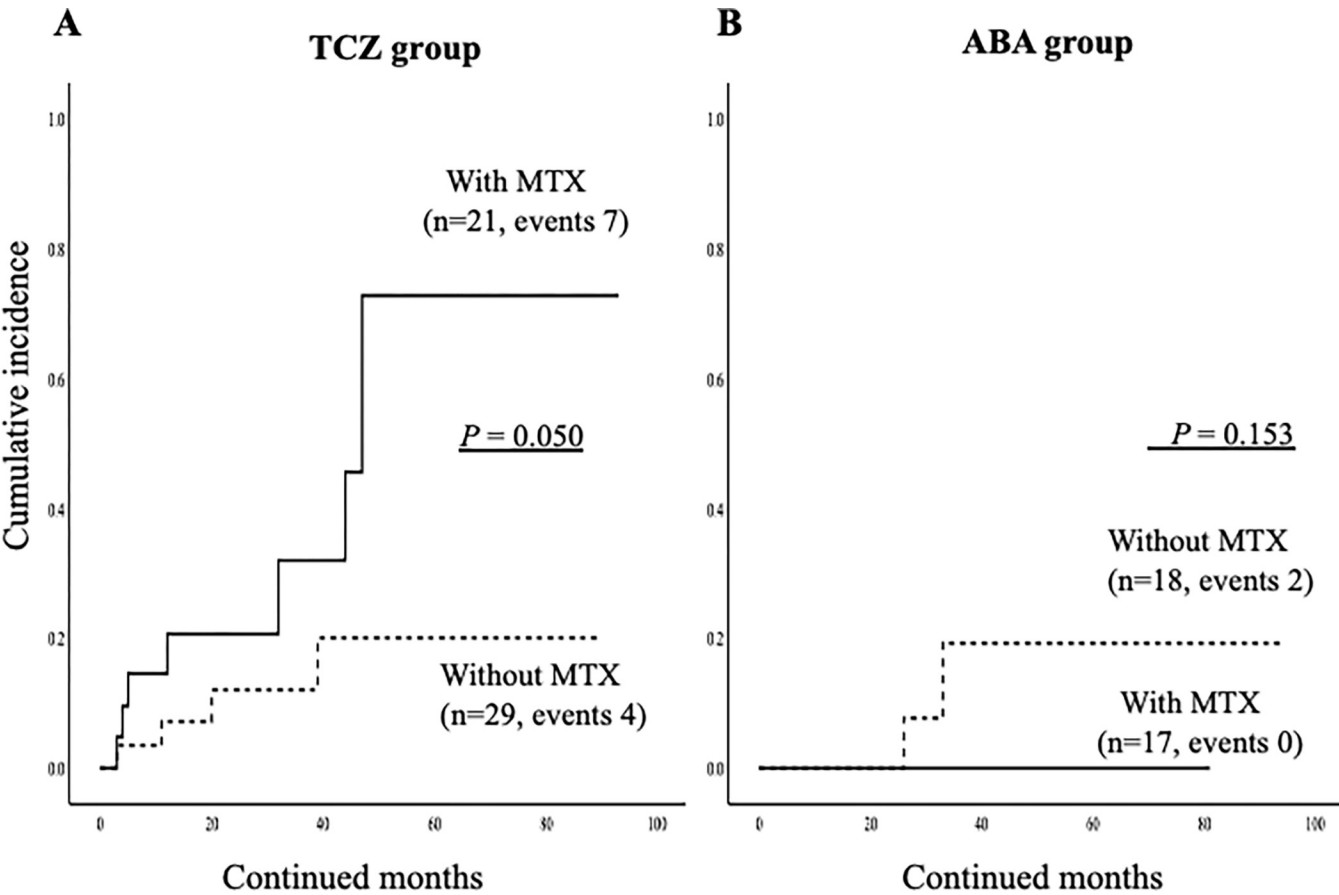

**Fig 10. The cumulative incidence of the discontinuation of ABT or TCZ due to AEs were compared in EORA group according to the concurrent MTX use.** (A) The incidences of the discontinuation of TCZ due to AEs were significantly higher in EORA patients with the use of MTX compared to those without the use of MTX. (B) There were no significant differences in the incidences of the discontinuation of ABT due to AEs between EORA patients group according to the concurrent MTX use.

of each rheumatologist, with no standardized protocol. Third, the sample size was relatively small. Fourth, marked reductions in the erythrocyte sedimentation rate and CRP levels were observed during TCZ treatment, which may or may not have corresponded to changes in other clinical signs and symptoms.

## Conclusions

The results of our study demonstrated that ABT and TCZ have similar clinical responses and drug retention rates in elderly patients in a real-world setting. Whereas, the rates of discontinuation due to AEs, including ILD, seem to be lower in elderly patients with RA taking ABT than in those taking TCZ. Furthermore, large-scale prospective studies are necessary to determine whether ABT exerts more protective effects against ILD than other bDMARDs in elderly patients with RA.

## Supporting information

**S1 Dataset.**
(XLSX)

## Acknowledgments

The authors are grateful to Enago (http://www.enago.jp) for the English language review.

## Author Contributions

**Conceptualization:** Kiyoshi Migita.

**Data curation:** Jumpei Temmoku, Eiji Suzuki, Yuya Sumichika, Kenji Saito, Shuhei Yoshida, Haruki Matsumoto, Yuya Fujita, Naoki Matsuoka, Shuzo Sato, Hiroshi Watanabe.

**Formal analysis:** Jumpei Temmoku, Haruki Matsumoto, Yuya Fujita, Tomoyuki Asano, Kiyoshi Migita.

**Funding acquisition:** Jumpei Temmoku, Haruki Matsumoto, Kiyoshi Migita.

**Methodology:** Jumpei Temmoku.

**Project administration:** Jumpei Temmoku.

**Resources:** Jumpei Temmoku.

**Software:** Jumpei Temmoku.

**Supervision:** Jumpei Temmoku, Masayuki Miyata, Kiyoshi Migita.

**Validation:** Jumpei Temmoku.

**Writing – original draft:** Jumpei Temmoku, Kiyoshi Migita.

**Writing – review & editing:** Kiyoshi Migita.

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
