## [Decision Letter · Decision Letter 0]

14 Jul 2022

PONE-D-22-17384Comparing the effectiveness and safety of Abatacept and Tocilizumab in elderly patients with rheumatoid arthritisPLOS ONE

Dear Dr. Migita,

Thank you for submitting your manuscript to PLOS ONE. After careful consideration, we feel that it has merit but does not fully meet PLOS ONE’s publication criteria as it currently stands. Therefore, we invite you to submit a revised version of the manuscript that addresses the points raised during the review process.

Our reviewers found some interests in this manuscript, but also pointed out a number of critical issues that require improvement or even amendment. I ask the authors to fully respond to all comments made by the reviewers in the revised manuscript.

We look forward to receiving your revised manuscript.

Kind regards,

Masataka Kuwana, MD, PhD

Academic Editor

PLOS ONE

Journal Requirements:

Reviewers' comments:

Reviewer's Responses to Questions

**Comments to the Author**

1. Is the manuscript technically sound, and do the data support the conclusions?

Reviewer #1: Partly

Reviewer #2: No

2. Has the statistical analysis been performed appropriately and rigorously? 

Reviewer #1: I Don't Know

Reviewer #2: Yes

3. Have the authors made all data underlying the findings in their manuscript fully available?

Reviewer #1: Yes

Reviewer #2: Yes

4. Is the manuscript presented in an intelligible fashion and written in standard English?

Reviewer #1: Yes

Reviewer #2: Yes

5. Review Comments to the Author

Reviewer #1: The authors compared the efficacy and safety between abatacept (ABT) and tocilizumab (TCZ) in elderly patients with rheumatoid arthritis (RA). The focusing point of this study is important and interesting in the era of ageing society. The authors concluded that drug retention rates were equivalent between ABT and TCZ, but the rates of discontinuation due to adverse event (AE) seemed to be lower with ABT than TCZ in elderly patients with RA. However, this study contains several serious concerns.

major

1. The authors should make a strict distinction between the terms of “Elderly-onset RA” and “Elderly RA”. These two terms refer to different population of RA (i.e the former refers to RA diagnosed at age of 65 or older, and the latter refers to patients who are currently 65 years or older regardless of the age of RA onset.) and which population should be targeted depends on the aim of the study, however, the authors use these terms confusingly. Therefore, the authors should review the design of the study and definition of patients depending on the aim of this study and revise the manuscript.

2. The authors showed all AEs leading to the discontinuation of ABT or TCZ in Table 2, but some AEs (ILD, liver dysfunction, renal dysfunction, cardiovascular disease, hypothyroidism) are generally unlikely to be associated with ABT or TCZ administration. Thus, it is questionable whether those AEs are due to these drug administrations and discontinuation of ABT or TCZ were necessary. Thus, the author should add explanations about detailed situation of each AE.

3. The authors highlighted the risk of ILD in the TCZ group, but it is not well verified or described. First, it is necessary to describe whether the ILDs were new occurrences after the initiation of TCZ or exacerbations of already diagnosed. Also, effect of MTX should be considered.

4. Regarding efficacy, the authors only showed achieving rate of low disease activity according to CDAI and DAS28-CRP at 24 weeks. When I read the method (P7L2-5), it seems that this did not include remission. It is desirable to analyze it including remission to evaluate effectiveness of the treatments. The authors should also show the longitudinal changes of score of CDAI and DAS28-CRP, proportion of the category of each disease activity (remission, low-, moderate-, high-disease activity), each component of composite measures, if possible.

5. Please describe summary of rate of reasons for drug discontinuation in ”Drug overall retention rates” section in P11.

6. AEs should be listed as a whole AE and then divided into AEs that led to drug discontinuation and AEs that did not.

7. The author excluded patients with no history of csDMARDs in Figure 1. What is the reason for excluding these patients?

8. The authors should add more detail clinical characteristics at the time of initiation of ABT or TCZ in Table 1. Each component of composited measures (SJC, TJC, PtGA, PGA), concomitant use of csDMARDs other than MTX, proportion of chronic kidney disease (eGFR<60), dose of glucocorticoid, types of used bDMARDs should be added.

9. Table 1 showed 12 patients with ABT group and 25 patients with TCZ group have ever used prior bDMARDs use. Is that mean are all prior bDMARDs TNF inhibitors? If not, it is necessary to describe the person who used both ABT and TCZ during the period.

Minor

1. Please correct abbreviations in the manuscript and figures. Abbreviations should be defined at first mention and used consistently thereafter, but it has not complied with frequently (ex. Adverse event (AE), low disease activity (LDA), abatacept (ABT, ABA) etc.).

2. “In the matched analysis,・・・・(P2L12, in abstract)” is not appropriate sentence because it may mislead that background were adjusted by using Propensity score matching etc.

3. In P8-9, check and correct the results for the rate of glucocorticoid use, as the results are opposite in the text and in Table 1.

4. In Table 3, the number of MTX users in the AE group is incorrect.

Reviewer #2: This study compared efficacy and safety of Abatacept and Tocilizumab in elderly patients with rheumatoid arthritis and showed comparable efficacy between both agents. On the other hand, the rates of discontinuation due to AEs, including ILD, seem to be lower with ABT than with TCZ in elderly patients with RA.

Major comments

The reviewer has a concern with the inclusion criteria.

#1 This study enrolled elderly (age ≥65 years) patients, irrespective of onset age with a mean of 7 years. Onset age is one of the important factors to determine clinical features including therapeutic responses. The reviewer is afraid that the authors confuse clinical features of elderly RA patients with those of elderly onset RA (EORA) patients. For example, the author described “Elderly patients with RA seem to have a characteristic pattern with a more acute onset, systemic involvement, and worse functional outcomes” (P19, L14-15), but this report compared EORA patients with young onset RA patients. The reviewer suggests analyzing the data by stratifying onset age or focusing those in patients having onset age over 60 years old.

#2 This study applied the 1987 ACR classification, but not 2010 ACR/EULAR, in spite of the study duration from 2014 to 2021. The reviewer is afraid that the inclusion criteria missed some of elderly onset RA patients. The reviewer suggests adding the data when the patients are enrolled based on the 2010 ACR/EULAR criteria.

Minor comments

#3 The reviewer suggests showing the administration routes and the standard dose of ABT and TCZ, because the standard dose of TCZ is different among countries. In addition, the interval of subcutaneous injection with TCZ can be shortened to one week. The reviewer is also interested in how often the interval is shortened because of the insufficient efficacy.

#4 Table 1 shows 25.5% of patients treated with ABT and 31.3% of those with TCZ have received prior bDMARD The reviewer is afraid it is hard to interpret the data in patients who had a switch from ABT to TCZ or TCZ to ABT. To avoid the complexity, the reviewer suggests showing the data in patients who received ABT or TCZ as the first bDMARD.

#5 spelling error

“the ACPA possibilities” (P17, L15) => positivity

6. PLOS authors have the option to publish the peer review history of their article (what does this mean?). If published, this will include your full peer review and any attached files.

Reviewer #1: No

Reviewer #2: **Yes: **Mitsuhiro Takeno

---

## [Author Response · Author response to Decision Letter 0]

28 Aug 2022

Reviewer #1: The authors compared the efficacy and safety between

abatacept (ABT) and tocilizumab (TCZ) in elderly patients with

rheumatoid arthritis (RA). The focusing point of this study is important

and interesting in the era of ageing society. The authors concluded that

drug retention rates were equivalent between ABT and TCZ, but the rates

of discontinuation due to adverse event (AE) seemed to be lower with ABT

than TCZ in elderly patients with RA. However, this study contains

several serious concerns.

major

1. The authors should make a strict distinction between the terms of “

Elderly-onset RA” and “Elderly RA”. These two terms refer to different

population of RA (i.e the former refers to RA diagnosed at age of 65 or

older, and the latter refers to patients who are currently 65 years or

older regardless of the age of RA onset.) and which population should be

targeted depends on the aim of the study, however, the authors use these

terms confusingly. Therefore, the authors should reviewthe design of the

study and definition of patients depending on the aim of this study and

revise the manuscript.

We appreciate your critical comments. According to your important comments, we discriminate “Elderly RA” and EORA and corrected these descriptions in the revived manuscript. We also performed the same analysis using the selected elderly-onset RA patients as requested by Reviewer #2. We presented these new data in the revised manuscript. 

2. The authors showed all AEs leading to the discontinuation of ABT or

TCZ in Table 2, but some AEs (ILD, liver dysfunction, renal dysfunction,

cardiovascular disease, hypothyroidism) are generally unlikely to be

associated with ABT or TCZ administration. Thus, it is questionable

whether those AEs are due to these drug administrations and

discontinuation of ABT or TCZ were necessary. Thus, the author should

add explanations about detailed situation of each AE.

We appreciate your critical comments. According to your important comments, we described the detailed situations concerting the discontinuation of TCZ or ABT due to AEs in the revised manuscript.

3. The authors highlighted the risk of ILD in the TCZ group, but it is

not well verified or described. First, it is necessary to describe

whether the ILDs were new occurrences after the initiation of TCZ or

exacerbations of already diagnosed. Also, effect of MTX should be

considered.

We appreciate your critical comments. According your comments, we described the detailed information concerning the occurrence of ILD and the effects of MTX in TCZ group in the revised manuscript.

4. Regarding efficacy, the authors only showed achieving rate of low

disease activity according to CDAI and DAS28-CRP at 24 weeks. When I read

the method (P7L2-5), it seems that this did not include remission. It is

desirable to analyze it including remission to evaluate effectiveness of

the treatments. The authors should also show the longitudinal changes of

score of CDAI and DAS28-CRP, proportion of the category of each disease

activity (remission, low-, moderate-, high-disease activity), each

component of composite measures, if possible.

We appreciate your critical comments. According to your precise comments, we presented these important data in the new Figure in the revised manuscript.

5. Please describe summary of rate of reasons for drug discontinuation

in ”Drug overall retention rates” section in P11.

According to your important comments, we described the rates of reasons for drug discontinuation in this section of the revised manuscript.

6. AEs should be listed as a whole AE and then divided into AEs that led

to drug discontinuation and AEs that did not.

According to your important comments, we listed whole AEs dividing into AEs leading to or not-leading to drug discontinuation in the revised Table 2.

7. The author excluded patients with no history of csDMARDs in Figure 1.

What is the reason for excluding these patients?

We appreciate your important comments.

These patients were excluded from the analysis due to the following reasons: 

Since the pre-existing serious complications (progressing ILD, renal insufficiency, hematological disorders, advanced disability), which limit the use of MTX and may affect the clinical course in these patents.

8. The authors should add more detail clinical characteristics at the time of initiation of ABT or TCZ in Table 1. Each component of composited measures (SJC, TJC, PtGA, PGA), concomitant use of csDMARDs other than MTX, proportion of chronic kidney disease (eGFR<60), dose of glucocorticoid, types of used bDMARDs should be added.

We appreciated your critical comments. According to your comments, we presented this important information in new Table 1 in the revised manuscript.

9. Table 1 showed 12 patients with ABT group and 25 patients with TCZ group have ever used prior bDMARDs use. Is that mean are all prior bDMARDs TNF inhibitors? If not, it is necessary to describe the person who used both ABT and TCZ during the period.

We appreciate your important comments. According to your comment, we added these important information in revised Table1.

Minor

1. Please correct abbreviations in the manuscript and figures.

Abbreviations should be defined at first mention and used consistently thereafter, but it has not complied with frequently (ex. Adverse event (AE), low disease activity (LDA), abatacept (ABT, ABA) etc.).

We corrected these abbreviations in the revised manuscript. 

2. “In the matched analysis,・・・・(P2L12, in abstract)” is not

appropriate sentence because it may mislead that background were

adjusted by using Propensity score matching etc.

We appreciate your critical comments. We deleted these sentences in the revised manuscript. 

3. In P8-9, check and correct the results for the rate of glucocorticoid

use, as the results are opposite in the text and in Table 1.

We corrected these mistakes in the revised manuscript.

4. In Table 3, the number of MTX users in the AE group is incorrect.

We corrected these mistakes in the revised manuscript.

Reviewer #2: This study compared efficacy and safety of Abatacept and

Tocilizumab in elderly patients with rheumatoid arthritis and showed

comparable efficacy between both agents. On the other hand, the rates of

discontinuation due to AEs, including ILD, seem to be lower with ABT

than with TCZ in elderly patients with RA.

Major comments

The reviewer has a concern with the inclusion criteria.

#1 This study enrolled elderly (age ≥65 years) patients, irrespective of

onset age with a mean of 7 years. Onset age is one of the important

factors to determine clinical features including therapeutic responses.

The reviewer is afraid that the authors confuse clinical features of

elderly RA patients with those of elderly onset RA (EORA)patients. For

example, the author described “Elderly patients with RA seem to have a

characteristic pattern with a more acute onset, systemic involvement,

and worse functional outcomes” (P19, L14-15), but this report compared

EORA patients with young onset RA patients. The reviewer suggests

analyzing the data by stratifying onset age or focusing those in

patients having onset age over 60 years old.

We appreciate your critical comments. According to your important comments, we analyzed the same analysis subjected RA patients with the onset age over 60 years old. We presented these new results in the revised manuscript.

#2 This study applied the 1987 ACR classification, but not 2010 ACR/

EULAR, in spite of the study duration from 2014 to 2021. The reviewer is

afraid that the inclusion criteria missed some of elderly onset RA

patients. The reviewer suggests adding the data when the patients are

enrolled based on the 2010 ACR/EULAR criteria.

We appreciate your critical comments. According to your comments, we applied the 2010 ACR/EULAR classification criteria, in the revised manuscript. However, we could not expand the number of the elderly RA patients.

Minor comments

#3 The reviewer suggests showing the administration routes and the

standard dose of ABT and TCZ, because the standard dose of TCZ is

different among countries. In addition, the interval of subcutaneous

injection with TCZ can be shortened to one week. The reviewer is also

interested in how often the interval is shortened because of the

insufficient efficacy.

We appreciate your critical comments. According to your comments, the administration ruts of ABT and TCZ in the revised manuscript.

#4 Table 1 shows 25.5% of patients treated with ABT and 31.3% of those

with TCZ have received prior Bdmard. The reviewer is afraid it is hard to

interpret the data in patients who had a switch from ABT to TCZ or TCZ

to ABT. To avoid the complexity, the reviewer suggests showing the data

in patients who received ABT or TCZ as the first bDMARD.

We appreciated your critical comments. According to your important comments, we added the important information concerning the prior Biologics in the revised Table 1.

#5 spelling error

“the ACPA possibilities” (P17, L15) => positivity

According to your precise comments, we corrected these mistakes in the revised manuscript.

---

## [Decision Letter · Decision Letter 1]

6 Sep 2022

Comparing the effectiveness and safety of Abatacept and Tocilizumab in elderly patients with rheumatoid arthritis

PONE-D-22-17384R1

Dear Dr. Migita,

We’re pleased to inform you that your manuscript has been judged scientifically suitable for publication and will be formally accepted for publication once it meets all outstanding technical requirements.

Kind regards,

Masataka Kuwana, MD, PhD

Academic Editor

PLOS ONE

Additional Editor Comments (optional):

Reviewers' comments:

Reviewer's Responses to Questions

**Comments to the Author**

1. If the authors have adequately addressed your comments raised in a previous round of review and you feel that this manuscript is now acceptable for publication, you may indicate that here to bypass the “Comments to the Author” section, enter your conflict of interest statement in the “Confidential to Editor” section, and submit your "Accept" recommendation.

Reviewer #1: All comments have been addressed

2. Is the manuscript technically sound, and do the data support the conclusions?

Reviewer #1: Yes

3. Has the statistical analysis been performed appropriately and rigorously? 

Reviewer #1: Yes

4. Have the authors made all data underlying the findings in their manuscript fully available?

Reviewer #1: Yes

5. Is the manuscript presented in an intelligible fashion and written in standard English?

Reviewer #1: Yes

6. Review Comments to the Author

Reviewer #1: The authors responded appropriately to reviewer's comments.

Please correct the abbreviation of abatacept in Figure 7 and 10 (ABA→ABT).

7. PLOS authors have the option to publish the peer review history of their article (what does this mean?). If published, this will include your full peer review and any attached files.

Reviewer #1: **Yes: **Satoshi Takanashi

---

## [Editor Report · Acceptance letter]

9 Sep 2022

PONE-D-22-17384R1 

Comparing the effectiveness and safety of Abatacept and Tocilizumab in elderly patients with rheumatoid arthritis 

Dear Dr. Migita:

I'm pleased to inform you that your manuscript has been deemed suitable for publication in PLOS ONE. Congratulations! Your manuscript is now with our production department. 

Kind regards, 

on behalf of

Prof. Masataka Kuwana 

Academic Editor

PLOS ONE